# A Design Methodology Using Prototyping Based on the Digital-Physical Models in the Architectural Design Process

**Do Young Kim**

Department Future Technology and Convergence Research, Korea Institute of Civil Engineering and Building Technology (KICT), Gyeonggi-do 10223, Korea; doyoungkim0123@kict.re.kr

**Abstract:** In this study, a design methodology based on prototyping is proposed. This design methodology is intended to enhance the functionality of the test, differentiating it from the prototyping that is being conducted in conventional architectural design projects. The objective of this study is to explore reference cases that enable designers to maximize the utilization of both digital models and physical models that have been currently used in architectural designs. Also, it is to explore the complementary roles and effects of digital models and physical models. Smart Building Envelopes (SBEs) are one of challenging topics in architectural design and requires innovative design process included tests and risk management. A conceptual prototyping-based model considering the topic is applied to the design studio (education environment in university). Designing SBEs is not difficult to conceive ideas, but it is impossible to "implement" using the conventional design method. Implementing SBEs requires to strengthen validities and improve responsibilities of ideas in the stages of architectural designs, with cutting-edge technologies and smart materials. The design methodology enables designers (represented by students) to apply materials and manufacturing methods using digital models (parametric design, simulation, BIM) and physical models, rather than representing vanity images that are considered simple science fiction.

**Keywords:** prototyping; digital model; physical model; smart building envelope; design process

## 1. Research Background

In response to the demands of the 4th Industrial Revolution [1,2], there is a need for a way to rapidly derive users' needs and strategies in the building design field [3]. On the other hand, in the automotive and aerospace fields, designs have been developed in response to the changes in the contemporary requirements, and the results have been achieved through a method called "prototyping" [4,5].

Prototyping can be explained by the meaning of "prototype." The word "prototype" is used in two ways: As an original model of something that serves as a basis for other things (Wikipedia, 19 February 2019) and as an "early" sample or model, including the functions of tests, which are created to find a design solution [6]. In the former definition, "prototype" refers to the beginning of the design, the basis, and the standard. In the latter definition, "prototype" leads to the designer's assuming-solving process in the course of all the design processes. Especially, in terms of the design process, the latter definition is closely related to the meaning of economic benefits. According to Acquisition Logistic Guide [7], in the last steps of the research and development process, not only does the cost of the project increase; the opportunity to reduce the costs also sharply decreases. That is, the design risks need to be eliminated at the beginning of the design phase, then the costs and resources can be reduced. The success of projects depends on the clarification of the requirements at an early stage and on rapidly eliminating the design risks.

Prototyping is one of the ways to reduce design errors and eliminate the failure factors in the initial design phase. The process involving the test-refinement-completion of designs using prototypes is called "prototyping" [4,5]. Prototypes are used to test if designs respond to their intended performances (e.g., "Will the design work as intended by the designers?"): According to ISO 15686, performance means "quality criteria for key characteristics at a certain point in time". Some of these criteria include energy efficiency, temperature and humidity, and the aesthetic and the interactions among the building and users that designers pursue. For instance, in performance testing, the following questions are posed: "Will the design meet the end user's requirements?" and "Will the design meet the required environmental performance?" Designers can use prototypes for learning, communication, integration, and milestone setting through the process of improving prototypes. The prototype design is refined to reduce the risk in the process of testing the prototype, or if constrained by the design conditions, the design may be newly born with a prototype (design creation), by constructing different concepts. The automotive and aerospace sectors have undergone an industrial revolution and have succeeded in upgrading new materials and technologies. This process will be explained below in terms of prototyping.

The Industrial Revolution needed corporate strategies and business models to incorporate new technologies. The automobile and aerospace fields have successfully overcome the challenges posed by such a revolution. Automobile and aerospace designs are actively developing core technologies to maintain their existing competitiveness and to preoccupy the future markets.

Model T was an early prototype (called with a platform product) in automotive design [8–10]. Ford's Model T began with the introduction of cars at a price lower than those of the expensive cars at the time. To design a low-priced model, the production method was changed (mass production, conveyor belt method), thereby making the design lighter, and the color types and shapes were simplified. Its shape and basic skeleton/ appearance were almost identical to those of the present cars. Even after its development, it had a long-term impact on the automotive design of other companies, as well as its successors.

On the other hand, the Wright Brothers' airplane was also an early prototype of various aircraft designs with fixed wings [11]. The Wright brothers were the first inventors to have built the aircraft structure in such a way as to allow speed and airflow control. The three-axis control allows pilots to control aircraft and balance the aerospace effectively. This method introduced the basic principles of maneuvering the future fixed-wing aerospace.

In the two aforementioned cases (Model T and the Wright brothers' aircraft), an important key is included concerning how prototyping is used to achieve breakthroughs in the design of automotive and aerospace applications [12]. In these two cases, the shape and structure of the design are implemented by representing design concepts and performance. This implementation was intended to provide a quick experience of testing and failure, and could lead to future improvements and enhancements of prototypes. Prototypes make designers acknowledge the design requirements that are easier to understand. It also clearly shows the designers' various assumptions. This is one of the most ways of making tests proceed rapidly and accurately.

Traditionally, prototyping is also used in architectural design processes. For example, there are wooden models of Filippo Brunelleschi and clay models of Michelangelo. These are shapes for explaining the ideas of a designer to construction workers [13,14]. There are also Gaudí's plaster models for unique purposes [13,15]. These physical prototypes are comparable to today's remarkable digital models of complicated or large-capacity forms. Even they require deep understanding and multi-disciplinary knowledge of building methods and materials.

In particular, Brunelleschi's prototype is the same as the test-based prototype described above. Brunelleschi's prototype was not simply a means of visualization, but a means of exploring the concept of architectural design morphologically. For example, the shape that Brunelleschi was trying to design had characteristics that made it difficult to build using the existing methods, and he was able to realize it in the process of designing new structures and methods, by using mathematical knowledge to overcome

the challenges involved. Brunelleschi's domes were difficult to build through traditional construction methods. Both a special construction process and special construction equipment were developed to make this type of construction possible. The results of the prototyping process implemented by Brunelleschi were innovative compared to the form of the era.

Prototypes of traditional architectures, such as Brunelleschi's work are similar to the prototypes of mechanical designs described in the previous section. The architects of the era are actually such experts like artists, scientists, mathematicians and carpenters. However, due to the division and specialization of fields in architecture, it becomes difficult to accurately estimate productivities considering manufacturing processes and materials in the stage of architectural design. Thus, it is difficult to expect holistic performances like traditional designs in the stage of architectural design. Therefore, the prototype of the current architectural design remains only as a result of the technical implementation of the architect and the completion of the result. The use of prototypes for testing is often delegated to other experts. While the failures and solutions recorded during the testing process will play a decisive role in technology development, architects have no choice, but to focus more on the way they describe others than experience based on failure and resolution. Architects, for example, inevitably spend a significant amount of time creating reports that summarize their skills and achievements or visual presentations that are as good as 3D images.

For the future building designs, differentiated prototyping processes are required according to the aforementioned cases. New forms and technological application are required to respond to the changes in the climate environment and the users' needs. Designs based on cutting-edge technologies refer to application processes or high-performance materials that are not well known in conventional building designs. For example, there is a large amount of complex new materials that are not utilized for the general construction process. Despite the many advantages of these new materials over other construction materials in terms of performance, firms use the existing materials, due to the risks posed by the new materials. The main reason that the new materials are not used is that their prices are not reasonable. In addition, there are few experts who understand how these new materials should be used, and there are also no domestic design standards or specifications related to such materials [16]. Therefore, new prototyping methods are needed to support numerous tests about new materials/technologies in the design phase. Rather than making guesses about unknown performances or just exploring individual design performances, design processes need to be facilitated to identify and supplement the inherent risks in the design.

## 2. Research Objectives and Scope

A new design methodology is proposed in this work to help architectural design firms respond to global movements.

The purpose of this methodology is to increase the productivity of architects who are fundamentally faced by architects. The reason for low construction productivity is related to high tolerance characteristics compared to other designs [17,18]. This tolerance could be solved at the construction or use stage, but it is to strike the type of architecture that needs to be more modern (e.g., climate-responsive, smart enclosures). In fact, the problem of tolerance is related to situations that do not take into account all the variables involved in the design phase, or the variables involved in the ongoing process. In the process of completing a typical building and a differentiated building, there is a high possibility that the opinions and satisfaction of the end user are not reflected, which causes astronomical economic loss.

In this paper, we focus on advanced cases of designing buildings as products. The designer quickly identifies errors that can occur during the design process by simultaneously considering the material and manufacturing process at the design stage. This means that early ideas can actually be created. Of course, purposing performance is not always able to be directly linked to productivity. The key in these cases is not only measuring performance as a way to achieve productivity, but also checking the measured values, both qualitatively and quantitatively, in a visible way. In conclusion, the purpose of this paper is to show that designers are performing in a visible way and to solve them.

In general, the function of a digital model is known to enable a human to implement a complex form that is hard to be realized by calculation or imagination. Parametric design, simulation tools, and building information models are among the digital technologies that are currently being used in large-scale, atypical projects in the construction sector. The design firms (Zaha Hadid, Herzog de Meuron, Gehry Technology, etc.) have been able to easily implement complex geometry with mathematical rules and use algorithms to analyze the performance of buildings to improve design accuracy and building performance. However, it should be noted that, despite the efficiency of the digital model, most of the experiments taking into account the properties of the material, characteristics of the site and surrounding environment, and manufacturing process are realized through the physical model. This aspect is also evident in the design training environment. Of course, the level of digital and physical models in practice and in the educational environment varies in size and in proximity to the end result. However, the complement of the two models proceeds at the same time without the internal validity of the design being established, not just the result. This clearly recognizes that the digital model's functions are responsible for new and convenient information processing and analysis, but emphasizes that the analog approach is visually appealing and familiar with the physical aspects of the site.

Despite the acknowledgment of the complementarity between digital and physical models, there is no example of defining the complementary role of both models, focused on productivity and performance efficiency. Therefore, in this paper, a design methodology is based on prototyping model using digital models and physical models to verify the effect of complementarity on designers. Of course, in the future digital models are able to substitute to physical models perfectly. However, in order to make highly functional and advanced products through digital technology, it is essential that the complementary utilization of two models is essential. It is because the physical model is an easy-to-make imagination to realize or validate plans according to the designer's intentions.

Suggesting a design method based on complementarity using digital models and physical models involves two purposes.

First, a design methodology that can minimize design risks and can create a timely response to the rapid changes in the climate environment and users' needs is necessary. The complementarity of digital and physical models (digital-physical models) theoretically minimizes the risks, which ultimately increases the reliability of the design. Digital and physical models used in the conventional designs are primarily employed to explore personal preferences like aesthetics of forms rather than focusing on the many design risks, such as environmental performance. In the design process of "Smart Building Envelopes (SBEs)", such use of digital and physical models is likely to lead to design failure. Unless many types of material, as well as manufacturing methods or environmental variables, are considered at an early stage, it will be impossible to realize the shape and structure according to the design intent. The failure of realizing the initial expected design leads to unbelievable economic losses, so differentiated design methods are essential.

Second, the design methodology using digital-physical models is essential for design groups that use technology convergence to carry out challenging tasks. According to the 4th Industrial Revolution, design firms are sympathetic to the need for a variety of technical attempts. Compared to other fields, however, the technical introductions in architectural design are remarkably slow. In addition, design firms using high technologies are generally reluctant to share their design data or outcomes. Of course, there have been some cases of SBEs, but they were not reproductions or generalizations of new SBEs. Most SBE designs are differently constructed or finally abandoned. This is because many design problems have not been recognized and solved in the design stages, and as such, the use of such SBEs results in unbelievable economic loss. The realization of smart envelopes requires a way to identify and resolve design risks immediately.

The purpose of this study is to apply the design methodology to the design practitioner. However, it is impossible for design practitioners to expect to participate in the experiment. The reason is that it takes time and spaces to experiment, and it is necessary to get financial support and cooperation in

advance. Due to this limitation, it should be replaced by a group similar to the design practitioner. As a similar group, a design studio, a kind of educational environment (novice designer) is adopted. The reason is because of the characteristics of current design studios and students. Although the importance of digital technology is still being addressed in today's design studios, it has been only theoretically possible to see how those roles work in the design process. This has been raised as a problem in the process of operating actual design studios. In addition, the role, and effects of digital technology are limited by the students' voluntary learning. We hope that another purpose of suggesting a methodology will be to help solve the gap between current teaching methods and the application of digital technology.

In other words, (1) a design model is proposed that utilizes parametric design, simulation, and building information model. It is suitable for one of the objects, such as the SBE based on existing advanced design examples (design firms and design studios); and (2) a design methodology is proposed based on student works and processes after experiments using the design model. The method of this proposal is based on the results of the two-year college students' participation (2nd semesters in 2015 and 2016—35 participants). Of course, there are insufficient quantities of cases to generalize with the methodology. However, readers can follow or utilize studied cases in implementations of innovative geometries without acknowledgment of the new method. Therefore, exploring the cases using complementary digital and physical models, prototyping-based models can be extended by applying new materials and technologies.

Of course, the educational environment (design studio) is not exactly the same as the working environment. This environment allows students to experience the design process indirectly through the guidance of the instructor who participated in the practice before they experienced the practice. Therefore, the educational environment is similar to the working environment.

First, through this case study, answers to specific questions are obtained. How do digital and physical models complement each other in the future? What are the effects on the complementarity of digital and physical models? With this content, practitioners attempting prototyping for the first time will be able to explore reference cases on the level and effectiveness of digital and physical models.

In this study, a design method is based on the complementary use of digital and physical models for designing SBEs. The design methodology using digital-physical models is focused on the advantages of existing tools: Digital tools, such as parametric design, simulation, and Building Information Modeling (BIM), and physical tools like wood and plastic board. This is a method that is based on the functions not only of digital models, but also of physical models. This is to take into account the situation in which physical models are actively used beyond the current developments of digital technologies. For example, it is true that digital technologies have evolved in such a way that digital models can now be implemented in virtual worlds, almost in the same manner as in the physical world. Design firms that create complex buildings or unique shapes also use physical and digital models. Physical models have the potential to stimulate and inspire designer ideas.

Indeed, design techniques like parametric design and simulation require knowledge of mathematics and of the data included in the form. While simple and small geometries may not be a big deal in digital modeling, it takes much time for complex design techniques to drastically improve the performances of designed forms or processes. As the number of building geometries and performances that a designer wishes to achieve increases, the optimization methods usually become more complex [19]. Misleading data and logic conforming to digital models may affect designers' choices [20]. Thus, physical models are essential to identify or straighten the imperfect or immature data extracted from digital models.

Proposed herein is a methodology for designing SBEs based on a process of prototyping using digital-physical models. The objectives of the work are to enable designers to embrace design implementation failures and learn from them early on. As illustrated in the framework, a case-based methodology is proposed. To derive examples based on the complementarity between digital and

physical models, conceptual models were formulated, and experiments were conducted on practical designers for two years (2015–2016) (Figure 1).

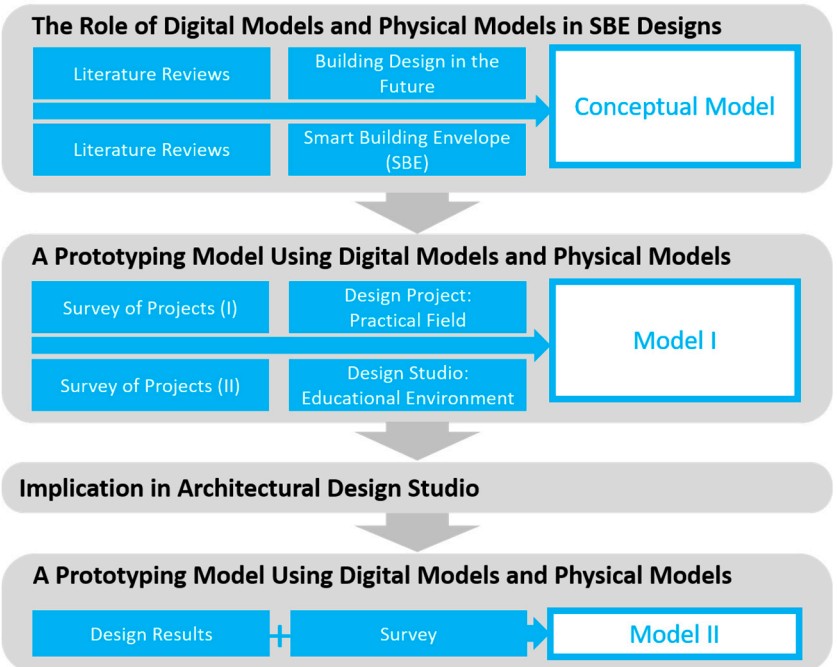

**Figure 1.** Research framework.

The research framework consisted of four parts, as discussed below.

First, a conceptual model of a prototyping model (①) was set up using the future building designs and SBEs. The existing literature was reviewed for the requirements for planning the prototyping model. The future building designs will focus on the roles of digital and physical models in strengthening the design process by considering the current design techniques, and SBEs focus on the parts of the design process that need to be particularly contributed by the digital and physical models.

Second, an initial prototyping model (②) was set up using cases of design projects carried out by design firms (Kieran Timberlake, Gehry Technologies, and Withworks). In such cases, both digital and physical models were used (complementary use) to develop a new construction method and smart materials. They were representative cases of efficient, accurate, reproducible, and secure designs because they were based on data and experiments. The successes of the three design projects that were used in this work depended on the conduct of early tests for finding the risk factors and early resolutions [21–25]. This is impossible in traditional design methods. A prototyping-based process model was established by analyzing the roles and functions of digital and physical models.

Third, the model was modified in detail through the project of lab-based design studios (③). "Design studio" is a method of operating classes, and "lab-based design studio" is a design studio based on collaborative work with supportive laboratories. The design studio is almost the same as a practical design firm because it is operated by architectural designers and associated experts assuming practical roles. Although design studios are almost the same as practical design firms, the roles and functions of digital and physical models are more clearly defined in relation to novice designers.

Fourth, experiments were conducted with a group of designers as participants, to verify the effectiveness of the prototype-based process model. The usefulness of the prototyping model was analyzed in terms of whether the roles of the digital and physical models contribute to the task in the process. Then the effects of the prototyping model were surveyed, such as whether designers sympathize with the roles of digital and physical models.

In order to minimize the designer's time to distort design results, design results were collected through pin-up announcements at fixed intervals (twice a week). If the presentation period is long, there is a possibility of distorting the intention or the concept of the design. This is done through qualitative research because the number of people involved in the design studio is small. (1) Observation method of the student is based on "researcher's non-participation" (observation using video camera). To prevent observers from recognizing the researcher's observing behavior or forming familiarity with the researcher, a video camera can be set up to allow students to freely discuss during the design process (pre-installed at the beginning of the design studio for students to adapt to the device). (2) We also conduct surveys and team interviews to investigate how students' perceptions of both models (digital and physical models) change.

Because it is case studies that are difficult to generalize, the method of qualitative research is appropriate. It focuses on the importance of the various behaviors and outputs derived from the process, rather than the outcomes created through the completion of the experiment. Therefore, qualitative research is appropriate for this purpose. Moreover, since the design is related to the creativity of design, creating a closed and regular environment can be a means to control creativity itself.

Ideally, architectural design firms need to cooperate in testing prototyping models, but the work environment is usually much influenced by the daily or ad-hoc tasks related to the existing projects. It is impossible to perform experiments targeting actual design firms without giving them economic support or compensation. Thus, the design education environment is appropriate for testing the prototyping models as a similar target. Ultimately, this study sought to maximize the benefits of digital and physical models by using design cases (student design results), and to provide a reference material that will enable designers to use digital tools and physical materials at the right time and in the right situations.

## 3. Conceptual Model

### 3.1. A Conceptual Model Using Contemporary Digital Models and Physical Models

The roles of digital and physical models have changed, and the functions of both models are now expanded Figure 2.

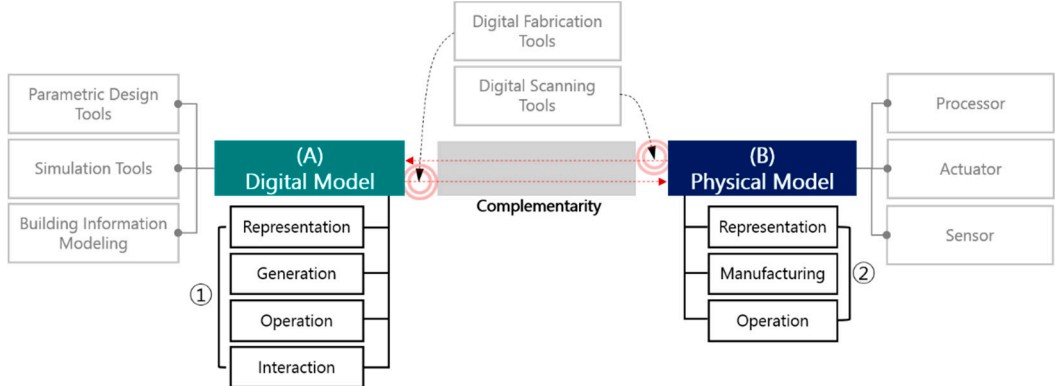

**Figure 2.** Prototyping using digital-physical models.

The paradigm of digital models and physical models allows designers to choose design solutions from as many alternatives as possible. In particular, by evaluating the design in terms of interactivity in terms of expression, finally, the optimized alternatives can be selected. The roles of the digital model have been extended to functions of representation, generation, operation, and optimization.

- Digital model

  - "Representation": Digital models like 3D models and 2D drawings have been extensively used in the form-making process. This involves iterative creation and modification of the operations requiring manual crafting by the designer as a computer operator.
  - "Generation" level: Due to the development of computational design tools like the generative parametric design based on scripts, digital models also better supports the regeneration of numerous offspring alternatives to help find the optimum forms [9,26].
  - "Operation" level: The integration of parametric techniques, integrated modeling methods (like BIM), simulation enables the operation of large amounts of building information, which makes it possible to generate alternatives in cost- and time-efficient ways rapidly. In particular, advanced computer simulation allows the calculation of the overall building performances (capabilities to respond to the designer's preference, end user's satisfaction, and environmental changes). Digital fabrication tools also enable designers to create a variety of geometric shapes, enabling designers to explore issues that are actually created or operated [27]. As such tools support the iterative process of generation-operation, the digital model changes from form-making to form-finding as designers take advantage of design solutions selected from among a myriad of possible alternatives.
  - "Interaction" level: The digital model using Virtual Reality, or Augmented Reality) helps implement the infinite functions of forms beyond the contemporary state of the form and the design operation [28]. If the digital models connect with physical models, their interactions would be changed according to the sensor data on the environmental conditions and user behaviors. Thus, it helps demonstrate the level of satisfactory performance for the end users in real time. This interaction between the digital and physical models involves various aspects pertaining to the use of all the aforementioned technologies. The current BIM and multi-dimensional realistic technologies like the virtual haptic and tangible space initiative have come to include this interaction. For example, BIM and the interactive augmented model have been expanded based on various sources, such as the terrestrial and airborne LIDAR, as well as other types of sensors. Finally, the digital model aids a prototyping process using the complementary digital and physical models by integrating 3D printing technologies and Internet of Things (IoT). This integration requires cumulative functions of the digital model, such as "representation," "generation", and "operation". Thus, the digital model supports the validation of prototypes by changing multiple ranges of operations and various types of shapes.

- Physical model

  - "Representation": On the other hand, the physical model is would according to the designer's intentions, the medium is strictly controlled by the material and the physical force (gravity, friction, etc.) [29]. Because they contain representations and tangibility of difficult to express using a digital model, texture, a physical model has received attention as a powerful medium of expression in the computerized age.
  - "Manufacturing": Digital fabrication technology allows physical models to be embodied in a form suitable for production in its original intended form [30,31].
  - "Operation": Sensors, actuators, and processors enable not only the physical model to be evaluated based on real-time data, but also the shape or movement thereof can be changed. For example, the shape, configuration, and characteristics of the elements constituting the physical model can be changed according to the user's information (name, occupancy, etc.) and environmental conditions (temperature, humidity, etc.) [32].

Smart building envelops (SBEs) is one of the challenging topics in architectural design and requires innovative design process included tests and risk management. The building envelope is a part of a building that is in contact with the interior or exterior of the building. It has a similar meaning as façade or a window system, and include the roofs of buildings. The SBEs' shapes or functions change in response to the surrounding environment (e.g., climate, indoor environment, or user requirements). Designing SBEs is not difficult to conceive ideas, but it is impossible to "implement" using the conventional design method. The SBE itself can be implemented using only digital or physical models, but it is difficult to fully validate unless prototyping-based processes are supported by these roles in each model.

## 3.2. A Conceptual Model Based on Prototyping

In the design of the SBE, complex requirements like structural reliability and operational safety, as well as design reliability and geometric constraints are required [33]. The design of SBEs is a process of coordinating the components' functionality in terms of integrated engineering as a process of finding the interactions between the components that meet the above requirements [34]. To satisfy these requirements, architectural design is required, as well as ICT, robotics, smart materials, and interaction design. SBEs are related to mechanical, electromechanical, material, and information technologies [34,35].

From the initial design stage of SBEs, designers must identify and resolve the errors that negatively impact the overall performance of buildings. Considering the geometry, system structure, and operating conditions, all together presents a problem that contributes to design failure. It is not easy to find the optimal movement and form of SBEs through the usual design process applied to general building products [33–41]. Based on examples of SBEs, design problems give rise to many issues, such as noise and operation failure: Maintenance cost (Arab World Institute), component lifetime (Novartis campus), power consumption (Bahrain WTC), and noise and vibration (Pearl River Tower). The economic loss is also required to be reduced by testing the shape, structure, operation, and manufacturing methods.

So far, technology and know-how have been accumulated for many years in the field of architectural design based on the uses of digital models (A) and physical models (B). The concept of the prototyping process is shown in Figure 3. It is set up to maximize the effects of current digital technologies.

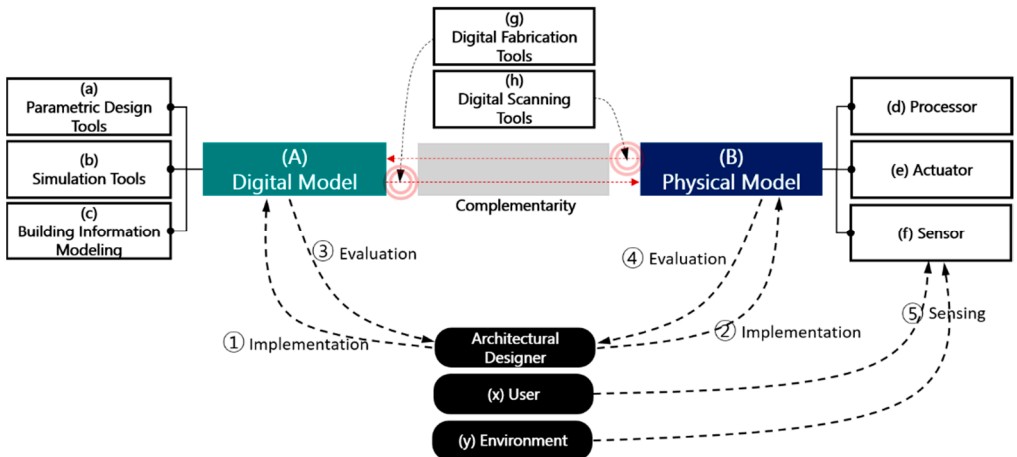

**Figure 3.** Prototyping process using digital and physical models in Smart Building Envelope (SBE) design.

The use of digital models can be explained through "parametric design" (a), "simulation" (b), "BIM" (c), and "digital fabrication" (d) (e.g., 3D printers and laser cutters). Parametric design (a) and BIM (c) allow creations and maintenances of shapes and motions based on the relationship between the components and the members. Simulation (b) also enables the evaluation of the design performance

without directly demonstrating the conditions of the surrounding environment. In addition, digital fabrication (g) makes it possible to quickly manufacture and operate complex shapes and components using real materials. By using these four functions, designers can easily find and construct reasonable shapes and systems.

The use of physical models can be explained through a "sensor" (f), an "actuator" (e), a "processor" (d), "digital fabrication" (g), and "digital scanning" (h). The sensor (f) is a tool that senses the positions/behaviors/conditions of the objects related to a specific task. The actuator (e) is a tool that drives the functions/parts of objects (e.g., parallel movement, rotation). The processor (d) converts the output of each device into meaningful information. It is necessary to analyze the information sensed by the sensor through the information processor, and to link the interpreted result to the act of the actuator. Digital fabrication (g) makes it possible to create shapes that are closer to reality than those created through 2D-based fabrication. It is important because it has made the sharing and production of digital files faster and reproducible. Digital scanning (h) helps designers read ready-made physical shapes. It is in fact, in the same context as the digital fabrication technology in terms of "digital file" and "rapid sharing."

The digital model allows the user to create a shape based on the rules defined by the designer (①). In addition, it can simulate the environmental performance and driving method of various alternatives and can help find the optimum design (②). Prototyping using the physical model allows the designer to test the shape and actual performance of the design through the implementation process (③). This helps identify problems with actual fabrication, such as the combination of materials and components (④). In addition, it is possible to represent the actual user's point of view in the process of acquiring data appropriate to the user's requirement, and to visually confirm the information of the actual environment (⑤). Both digital fabrication and digital scanning allow designers to reduce the design time and cost remarkably and to improve the complementarity of the digital and physical models implemented in non-matching.

Digital models help designers quantify the performance of designs across common test conditions, even without making physical models, but they do not represent all the various factors involved in designers' inspirations. It also does not represent physical conditions like the end user's sense, perception, cognition, and behavior. On the other hand, physical models help designers conduct a series of manufacturing processes in real environments. Using physical models, designers are able to gain real-time feedback to resolve design problems based on unexpected defects. To modify designs, however, it is necessary to prepare suitable materials, but also to secure extra time for the manufacturing processes, such as the fabrication and alignment of the components.

The point is how the usability of digital and physical models is maximized. The maximized functions of digital and physical models is an ideal case. To improve the usability of the two models, the role of the complementarity between them is essential. The next chapter shows the role of complementarity, which is analyzed using cases of architectural design projects.

*3.3. Case Studies of Architectural Design Firms*

Prototyping in the automotive and aerospace fields aims not only to explore ways of implementing geometries suitable for design concepts, but also to minimize the risks posed by the use of certain materials and techniques at the levels of the actual implementation. In fact, in architectural design, there are reasons for the necessity of conducting this research. Advanced digital technologies have been applied in the field of architecture design. The required time for the commercialization of digital technology, however, is somewhat long compared to the engineering field. This is because architectural design firms are reluctant to introduce new materials and technologies, due to the economic burdens involved and their lack of understanding of materials and technology. Of course, advanced architectural design firms (e.g., Kieran Timberlake, Gehry Technologies, and Withworks) have applied new types of materials and techniques in architectural design. They have made digital and physical mock-ups from the initial levels of the design processes.

Studies dealing with both the digital and physical models are categorized into two types. They deal with complementarity as an "alternative" aspect [42] or a "synergy" aspect [43,44]. Studies on complementarity as an "alternative" aspect emphasize the fact that the roles of digital models increasingly replace the roles of physical models. Studies on complementarity as a "synergy" aspect, on the other hand, emphasize the fact that despite the development of digital models, physical models have unique functions that cannot be replaced by digital models. The previous "synergy" researches, however, have not clearly explained "complementarity" (the relationship between digital and physical models). Proposing cases of "complementarity" helps designers not only test actual designs easily than test using only digital or physical models, but also find design inspirations. Thus, case studies are required to entice more designers to conduct design processes using digital-physical models. Assuming that designers utilize cases of prototyping process using digital-physical models, when they try to realize complicated geometries, prototyping enables them to create complicated geometric shapes in digital models. It also enables them to learn and apply the shape deformation techniques in physical models without trying to implement them in digital models.

The roles of digital and physical models were analyzed through design projects. Companies like those aforementioned have used digital and physical models in the design phase (Table 1).

**Table 1.** Roles of digital-physical models in architectural projects.

| Name of Design Firm | Digital Model | Physical Model |
|---|---|---|
| **Kieran Timberlake** | ■ Planning the envelope pattern<br>■ Planning the structure<br>■ Planning for the combinations and details of the members<br>■ Planning space and modular systems<br>■ Planning and simulating construction<br>■ Simulating the details and combinations | ■ Combining smart and ready-made materials<br>■ Realizing the performances of smart materials<br>■ Implementing the production process and method<br>■ Testing the combinations and details of the members<br>■ Testing the production of panels<br>■ Establishing a pilot environment considering the site environment<br>■ Testing the real-time performance |
| **Gehry Technologies** | ■ Exploring shapes with structures, members, and divided panels<br>■ Analyzing or reviewing the performance of the geometry<br>■ Reviewing the effects of a series of construction processes and construction methods<br>■ Reviewing the details (joint part shape)<br>■ Reviewing the construction method with the details | ■ Making curved panels<br>■ Reviewing the performance in a real environment<br>■ Reviewing the method of heat-treating glass<br>■ Conducting tests through the shape of the joints and the welding process<br>■ Testing the combinations of the members in the structure (beams)<br>■ Testing the combinations of the sublayers of multiple layers<br>■ Reviewing the construction tolerance |
| **Withworks** | ■ Mathematically analyzing the shape<br>■ Reconstructing the shape based on the production<br>■ Planning and analyzing the details considering the features of the shape<br>■ Planning a system of structures<br>■ Planning a panel system based on construction<br>■ Reviewing the construction tolerance | ■ Changing and examining the overall shape according to the change in the construction method used<br>■ Applying and testing materials, examining the feasibility of mixing with existing methods<br>■ Testing the safety<br>■ Testing the accuracy of construction and production |
| **Categories of tests** | ■ Form (FO): External shapes, internal shapes (space), details<br>■ Material (MA): Properties of the material<br>■ Operation (OP): Movements and facilities<br>■ Fabrication (FA): Production method | |

Kieran Timberlake has implemented smart-façade and modular building systems that incorporate prefabrication and embedded technologies. The success of design projects depends on the development of joints and the fabrication method used. Design projects have been operated with completed new design methods, such as the standardization of assemblies and the mixing of general and smart



materials. Withworks has optimized free-form systems while studying a new design method using existing technologies. To realize the shapes of free forms, the problems in the structural details and construction technologies need to be solved to make up for the disadvantages of the existing construction materials. The problems regarding the shapes have no correct solutions, and the success of the use of shapes depends on the time and cost. Gehry Technologies has realized Gehry's ideas. To implement a unique geometry using general materials and glass, it has also developed construction methods to fabricate optimized panel surfaces using unique software and field experiments.

Kieran Timberlake sought to explore building systems using embedded technologies like smart materials, information processing, and material mechanics. There has been no reference that allows the integration and output of multiple layers of materials, while keeping the thickness of a single substrate thin. Through prototyping, however, layers of general materials have been integrated with polyethylene terephthalate (PET). As devices or materials based on technologies are not commonly applied at the design stages, the firm needs to experiment in real environments, without reference cases for a single system. They call this work "science."

Gehry Technologies starts the initial programming and design phases by scanning the physical model in three dimensions. This is done to digitize the physical model for the development of the technical part or form. The row data acquired through physical models are structured through professional software, and the simulations for the environmental changes and production are used in the initial stage of the design process.

Withworks has proceeded to make molds of irregular shapes and to optimize panels. There are two matters, however, that need to be overcome: The domestic steel engineering specialists are not familiar with the irregular concrete casting and the optimization of the panel, and the curvatures of free-form design are not constant. Designs are, thus, required to transform the curvatures through optimization, and to change the methods of casting irregular concrete in the field. Particularly, digital models are used as a means to evaluate construction errors using the scanned data extracted from physical prototypes. Physical prototypes are called "visual mock-up." They are differentiated from general physical models because they can be digitized as data.

What is common among the three aforementioned companies is that their designers have experienced a series of test-fails using the data of shapes with materials in the design stage. By making and studying designs in digital environments and in the physical fields, the firms have tried to maintain the intended shapes.

## 4. Course Design Using the Prototyping-Based Process

### 4.1. Case studies of Architecture Design Studios

In the case of design studios, there are lab-based design studios with laboratories that use advanced technologies (Table 2). Such laboratories study research methods using manufacturing technologies and new materials. They have also evolved to operate studios by applying digital and physical models, like practical firms of architectural design, based on the annual students' results.

According to the design results of the design studio cases, the functions of digital and physical models need to be extended. Because the functions originate from the characteristics and limitations of novice designers: Novice designers are explained through an opposite word as "experts". They are not only naïve about design process and the viability of products, but also have less interaction experience and relatively poor to use or obtain knowledge to get design goals. The digital and physical models in design studios are different from the cases of practical design firms, as follows.

The initial digital models are usually tantamount to the design principles for making design shapes (parametric models) and for the fabrication and transformation methods (simulation, BIM). As students are often either ignorant or have inaccurate knowledge of modeling or design shapes, they have difficulty implementing digital models. By making intermediate designs or parts of designs instead of the intended shapes, they try to control the digital models.

**Table 2.** Case studies of architectural design projects.

| Design Studio | Digital Model | Physical Model |
|---|---|---|
| **TU Delft, Bartlett, Carneige Mellon** | ■ Representing the type in actual construction (using parametric design tools)<br>■ Complex rule geometry representation (using parametric design tools)<br>■ Shape change expression (using simulation tools)<br>■ Prediction of shape change based on environmental variables (using simulation tools) | ■ Conducting performance evaluation of the shape at the concept phase<br>■ Determining if the shape can be constructed when the material is introduced<br>■ Trial implementation of the production process considering a new production method<br>■ Learning about techniques of introducing new production methods<br>■ Implementation of the practical production process incorporating production methods and properties |
| **Categories of Tests** | ■ Form (FO): External shapes, internal shapes (space), details<br>■ Material (MA): Properties of materials<br>■ Operation (OP): Movements and facilities<br>■ Fabrication (FA): Production method<br>■ Mid-tests (m-T): Intermediate trials and errors (★) | |

Note: In particular, novice designers mostly conduct a considerable amount of intermediate tests (★) for verifying uncertain information or hypotheses.

Physical models are used to gauge only the physical properties of materials that are difficult to examine using digital models. When representing or reviewing the combination of components, students need to consider expert knowledge of the final products in engineering. Physical models sometimes require designers to understand professional techniques like kinetic parts' operation mechanisms or data processing logics. For example, production methods need to be practiced first to find a suitable production method when considering forms and structures. Physical models are used to roughly add details or supportive parts to assist structures. Although full-scale models are important for accurately evaluating the design performance, physical prototypes are preferred to be implemented on a small scale. Small-scale models are used like blob prototypes (unstructured form), due to economic burdens.

For the prototyping process to reflect the users' characteristics, digital and physical models need to evolve a series of challenging-evaluation-learning unknown areas of material properties and fabrication methods.

*4.2. Course Design Based on Case Studies*

In the initial design phase, there are many factors concerning the building characteristics or components that are not determined. In the process of applying the unidentified parts sequentially, the digital and physical models may not be implemented at the same time. When modifications of each model occur, they affect the others. As there is familiarity with the existing teaching methods of design studios, however, examples should be presented to remind them of certain ideas. Cases of tools related to designing SBEs are shown: Referential digital tools like Rhino/Revit and Rhino plug-in (Figure 4, ①), and referential physical tools like Arduino (⑤). Cases of advanced design are also shown: The process and results of SBEs, automobiles, and aerospace (②). In the course of utilizing the physical model, a presentation method related to such a model is provided as an example so that the performance of the model can be viewed objectively (④). The reason for the offer is the preliminary questionnaire conducted among the students. In the surveys, the students are largely aware of the functions of digital models, and define these as "objective" and "rational" evaluation. On the other hand, they are not at all aware of the functions of physical models, and analyses using them focus on "visible" and "subjective"

evaluation. In addition, the basic manuals and use cases of the laser cutter are provided to enable it to be used as an intermediary medium to sequentially complement digital and physical models (③).

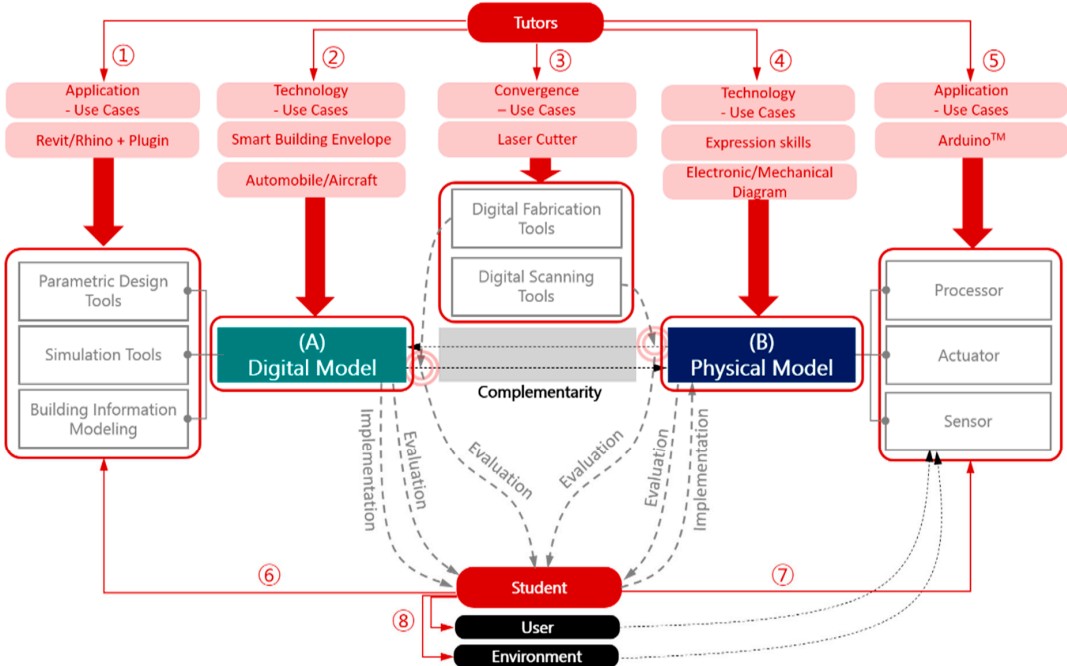

**Figure 4.** Prototyping using digital-physical models.

The design studio is operated with the theme of "kinetic facade." The kinetic envelope is a smart envelope, which began with passive movement, but its implementation features and performance vary according to its application to a wide range of advanced technologies, such as object networks, artificial intelligence, and smart materials. The applications of a broad range of technological applications are hard to apply in classes. The reason for adopting "kinetic facade" in this research was to allow students to focus on the transformations of forms and structures rather than on the complexities of technologies: The movements of the kinetic envelope require designers to visually orchestrate between components and to make clear structures, due to the smooth motion or function changes. It can be completed only through a perfect engineering-design combination.

Kinetic designs have greater limitations in meeting the intended performances. As "kinetic" means to respond to changes in shapes or functions, the designs are dealt with in real environments like frictions and weight loads, due to the movement of the components. They can also cause noise. They require a series of challenge-evaluation-refinement because designs are difficult to implement only by using the usual form and structure of the general envelopes, and they allow designers to attempt to replace commonly used materials with new materials, or to study methods of manufacturing materials.

The course for the design studio is largely categorized as a course for facilitating students' development of the design by themselves, and as an evaluation course by tutors who play the role of clients or end users. Students first extract abstract shapes or the intentions of the kinetic facade (①, ②, and ③ at level 1 in Figure 5). They also make models of components with various logics of shapes and motions in a digital model, or roughly combine the components that enable imagining the system's behaviors in physical models (④ at level 2). Designers create digital models of the offspring types based on the same schemas (using parametric designs to control the variables and creations of variations), or artificially manipulate movements, while maintaining the overall shapes of physical models (⑤ at level 3). Designers evaluate the performance of the design (⑥ at level 5) and vary the shapes and movements based on the current ideas (④and ⑤ at level 4), or if the current idea needs to be changed, an attempt is made to improve the performance in a different way, by creating a new prototype (⑦ at level 6).

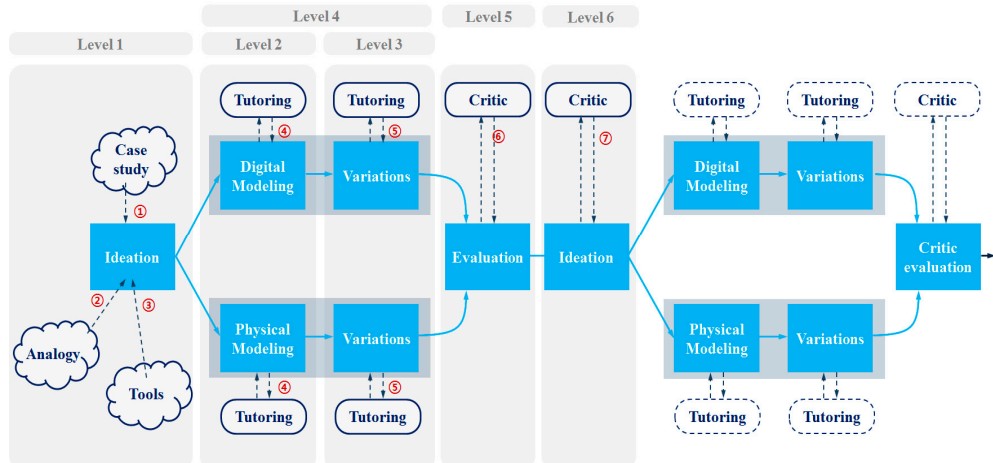

**Figure 5.** Course design based on prototyping using digital-physical models.

## 5. Verification

### 5.1. Design Results

Students are ideally able to predict performances by considering forms (FO), materials (MA), operation (OP), and fabrication (FA) using digital prototypes and physical prototypes. Digital prototypes, however, were used primarily in the process of implementing forms while assuming the material and the operation and fabrication methods. This is because students have difficulty implementing these in digital models, or time is wasted on the implementation. When students only assumed these problems (FO, MA, OP, FA) through such digital models, it was necessary to create a physical model that included the remaining problems.

Physical prototypes were used mainly to actualize material properties and fabrication methods, which are mostly unknown. For example, they were used to judge if they had structurally safe forms/movements against gravity, or if there were inappropriate frictions between components, or noise. When physical prototypes were used to implement a form, however, they were manufactured by digital prototypes. This was the reason for considering the exact dimensions and areas of components. To utilize physical prototypes to change designs' motions or functions by considering the surrounding environment, the use of digital devices was essential. By monitoring the data values of specific sensors or actuators, students were able to gauge the performance of small movements or the functional changes of prototypes, and to compare prototypes.

Even so, it was difficult to perfectly synchronize the digital and physical models. To come up with a perfect match between the digital and physical models, much content was needed, but it was not at this level. In fact, as many tests and selections had to be done initially to determine ideas, the process of mutually matching prototype types and functions was impossible and premature.

Table 3 shows an example of the complement between digital and physical prototypes. Each example of digital and physical prototypes contains a number of assumptions and conclusions. The previous and later prototypes in each example were compared in terms of FO, MA, OP, and FA. Each model has a different expression and assumption, even though it has similar contents to other models. The case of EX 1 can be explained as below. The digital prototype is a two-dimensional representation, which was considered a form that did not actually represent materials and operations, particularly how they were made. In this case, the physical prototype is a three-dimensional shape, but a conceptual form about the actual operation and manufacturing methods were just implied. In the case of EX 2, even if the previous physical prototype represented all the four aspects, the subsequent digital prototypes may contain incomplete information. This is because the physical prototype implementation did not meet the intended performance, and it was necessary to create a new shape or to fine-tune how it worked.

**Table 3.** Categories of design results.

| Num. | Images of digital models | | | | Dir. | Images of physical models | | | |
| --- | --- | --- | --- | --- | --- | --- | --- | --- | --- |
| | Functions | | | | | Functions | | | |
| EX1 |  | | | | ← |  | | | |
| | FO | MA | OP | FA | | FO | MA | OP | FA |
| | 0 | ★ | ★ | ★ | | 0 | ★ | ★ | ★ |
| EX2 |  | | | | ← |  | | | |
| | FO | MA | OP | FA | | FO | MA | OP | FA |
| | 0 | ★ | ★ | 0 | | 0 | 0 | 0 | 0 |
| EX3 |  | | | | → |  | | | |
| | FO | MA | OP | FA | | FO | MA | OP | FA |
| | 0 | ★ | ★ | ★ | | 0 | 0 | 0 | 0 |
| EX4 |  | | | | → |  | | | |
| | FO | MA | OP | FA | | FO | MA | OP | FA |
| | 0 | ★ | 0 | 0 | | 0 | 0 | 0 | 0 |
| EX 5 |  | | | | ← |  | | | |
| | FO | MA | OP | FA | | FO | MA | OP | FA |
| | 0 | ★ | ★ | ★ | | 0 | 0 | 0 | 0 |
| EX 6 |  | | | | ← |  | | | |
| | FO | MA | OP | FA | | FO | MA | OP | FA |
| | 0 | ★ | 0 | 0 | | 0 | 0 | ★ | 0 |

Note: O: Included; -: Not included; ★: Not included but assumed.

Digital and physical models are used to solve design problems through student work results. For example, the digital and physical models are largely utilized in three ways. First, it is used to reinforce the validity of ideas. In the early stages, designers plan for shape, performance, material composition, and data usage and explore existing methods and available resources to realize them. This step involves measurements, evaluations and re-measurements similar to the next step. However, there are various levels of implementation, failure, and learning because they explore various factors without knowing whether their intentions are right. Second, it is used to increase the reliability of ideas. Identify design strengths, find ways to improve weaknesses, and voluntarily find, measure, and measure existing planning and design methods that can contribute to real-world design. Third, it is used to refine ideas in detail. Look for ways to improve performance on detail, other than the main content confirmed in the second step.

*5.2. Surveys and Video Recording*

The questionnaire was administered to the students who participated in the class at the beginning and end of the curriculum. Most of the students had no experience in studying and participated in architectural design competitions. The first questionnaire (administered at the start of the curriculum) investigated "the experiences of measuring and evaluating design performance in the design process" in existing design studios. Specifically, the questionnaire was designed to predict what the students would gain from this study. In other words, the purpose of this study was to judge if the students usually agree with the functions of the digital and physical models, and with the complementary effect.

According to the results of the first questionnaire survey, the students had tried to come up with "eco-friendly" designs in their previous design studios. They did not verify the performances, however, using simulation or modeling, and just completed the design at the level of imagination or assumption. At the conceptual design stage, digital models had often been used to produce results or explore aesthetics rather than to verify or test something. Physical models had also been used as a means of presenting results.

Based on the results of the second questionnaire survey, some changes were found compared to the first questionnaire survey. First, the ranking of the factors that contribute to the design development changed (factors: Environmental performance, assessment of end user, feasibility, operational inefficiency, and designer's preference). At the start of the design project, the students were less concerned with the factors other than the designer's preferences, which are typically very important as design factors in the engineering field. At the end of the design project, however, the students changed their rankings. For example, they realized that objective data and physical phenomena in the real world were important. This was because the viability of designs is directly connected with objective performances like an assessment of users, evaluation of environmental performances, the inefficiency of operations, and cost (Figure 6).

The uses of digital and physical models have also changed compared to the early days of the project. For example, digital models were mainly used for visualizing creative forms during the presentation time. With the experience of the projects, digital models have been used to clarify the concept or express the validity of forms and structures. Although digital models were used as a visible method, they were different because their data or images were media for strengthening the students' ideas. For example, digital models were used as media for exploring parametric forms. Most of the students, however, gave up establishing the logics of the shape. For this reason, searching for various parametric shapes was not widely utilized as expected (Figure 7).

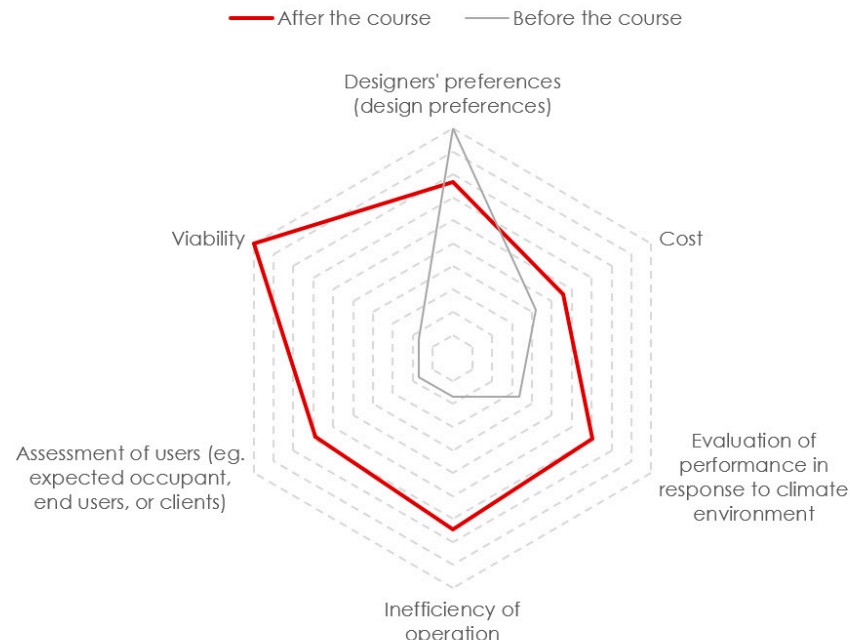

**Figure 6.** Factors that are important to the designers in evaluating the building.

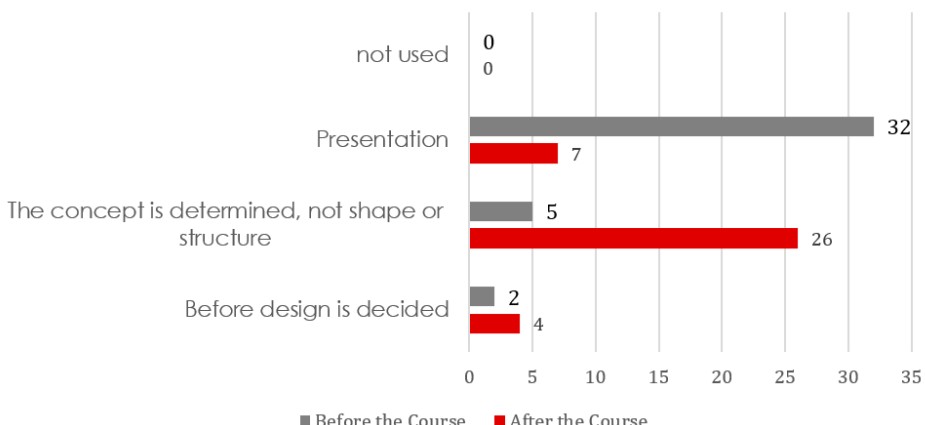

**Figure 7.** Functions of digital models in designing SBEs.

The physical model played other important roles and had common uses. In addition to the existing visual representation methods, it was used to investigate and evaluate performance using real models. What is unusual is that unlike digital models, which are known to be convenient, physical models have been conveniently handled to show and explain ideas to others (Figure 8).

Above all, what was most challenging for the students was the process of implementing envelope movement and working principles. In particular, communication with the team members and collaborating tutors are given priority over knowledge and theories when measuring and improving the performances of kinetic designs. In other words, when students implement design concepts, the material properties and manufacturing methods are the most important, with the students believing that the success of their design lies in quickly finding and solving the cause of failure.

Video recording allows the observer (researcher) to view multiple discussion processes within the team during the same period. For example, the recorded process includes the following: How to define concepts using a kind of performance, how to measure and visualize performance, and finally, whether the problem is clearly resolved. Because there were many types of unforeseen trial and error in this process, there were big and small discussions with team members and ideas. Of course, not all of the expected data were collected. As well as discussions of design concepts, discussions during

manufacturing and operations were expected to be recorded. Originally, students were expected to familiarize themselves with the video recording environment through a trial installation of one to two weeks. However, designers have not tried to manufacture and operate design models in video recording environments. The reason for this is the familiarity of the work environment. The space in which students make or operate the model was not a video recording environment, but a private workspace. In order to investigate behaviors of designers in the future, requirements of the design workspace are needed.

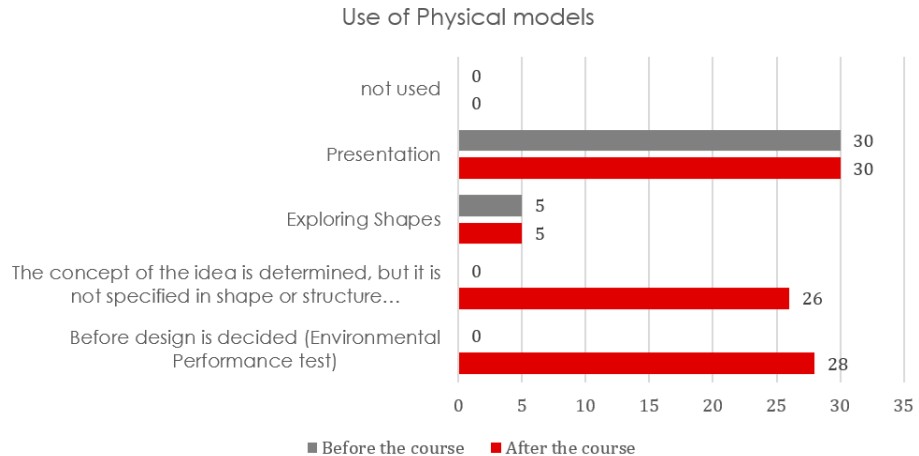

**Figure 8.** Functions of physical models in designing SBEs.

*5.3. Design Methodology Based on Prototyping*

In this prototyping process, the digital and physical models are designed in such a way that the designer does not rely on subjective tastes or assumptions, but has experienced a way of rationally improving the design through the evidence, such as quantitative data and phenomena.

In the design process, digital and physical models are used as a vehicle for identifying design problems and finding solutions to these (Figure 9). In the prototyping process, the complementarity between the digital and physical models depends on the level of "Strengthen validities of ideas" (Figure—①), "Improve responsibilities of ideas" (②), "Refine ideas" (③). The design process is divided into these three stages, and the roles of the digital and physical models are coordinated in each stage.

① Strengthen validities of ideas The designer explores the design concept to achieve his or her intended outcome. The initial concept is either the designer's pursuit of an ideal goal, untested based on the designer's experience or imagination. Designers use similar examples to illustrate the concept of the initial design, or similar cases where the intended outcome is similar to the case of the project. In this process, the complementarity between the digital and physical models enables the exploration of the method of implementing the characteristics and performance of the case equally. It also helps designers understand the principles of materials and techniques that are unfamiliar, and the procedures of using them. The design derived from the exploration of the case becomes clearer through experiments on the physical, mechanical, and chemical properties of the material and how it is used. Experiments are conducted under a variety of conditions to ensure that the theoretical part of the material and technology is known, and that the designer's hypothesis based on it is in fact fit. In this process, the mutual complementarity of the digital and physical models improves the understanding of the principle of the material and technology, and how to use it, and makes it possible to sample objects based on this.

② Improve responsibilities of ideas The designer explores how to shape design ideas in form, composition, and structure, with ideas for form and function determined. Designers combine the shape, color, and key elements of a building, and coordinate the function and performance of each element based on ideas. In this process, the complementarity between the digital and physical

models enables the comparison and evaluation of the aesthetic performance, the performance considering the environmental conditions, and the performance considering the user's needs. In addition, by considering the actual performance, such as the changes in the fabrication method and behavior, the design problems associated with the construction and maintenance stages can be addressed.

③　Refine ideas The designer explores ways of efficiently improving the creation and operation of shapes in a rough form and with the function determined. Even if the main forms and structures of the design have been determined, the process will continue to improve the design in detail. In this process, the complement of the digital and physical models is used to collect and process data, so that it can analyze the performance in detail. It also allows the thickness or height of the members to be adjusted, or the details to be adjusted as well to make up for the specific problems of the design.

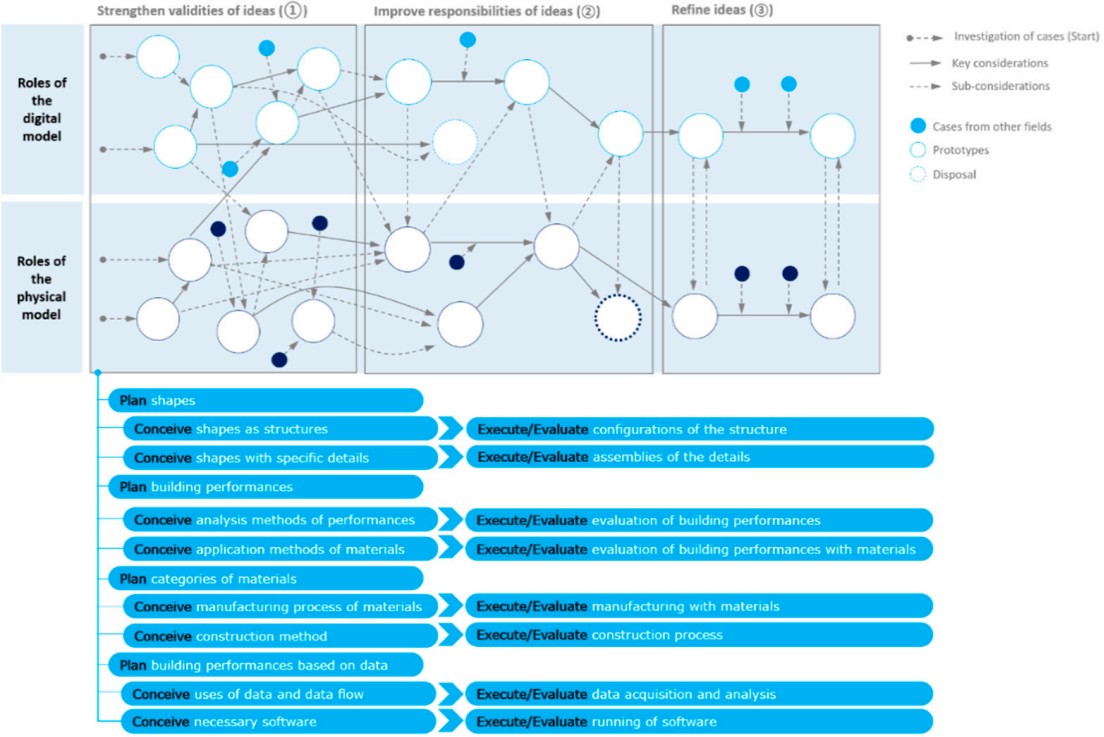

**Figure 9.** Design methodology using digital models and physical models.

## 6. Conclusions

It is important to specify the processes that use the digital and physical models, the performance of the product, and how to determine if the product is as effective as the designer intended it to be. The prototyping process is a step-by-step process of overcoming the design failure factors related to shape, structure, and operation according to design concepts. The process is important because the design failures are potentially related to the quality of design outputs. There are two types of failures that are addressed through the prototyping process.

First, it is possible to define the causes or characteristics of failure factors, but it is difficult to quantify or measure the potential loss and damage factors accurately. For example, the designer performs energy performance evaluation in the process of designing the building envelope. Through voluntary information retrieval and existing knowledge, it is understood that the energy performance of the building envelope is affected by environmental conditions like airflow, sunshine, and topography. The actual physical parameters related to the performance of the building envelope, however, vary. Simulations based on quantified formulas can make it difficult to find a definitive solution to the

problems that occur in the physical environment, especially if they include materials or technologies that have not been specifically applied.

Second, the failure factors are interpreted differently, or the degree of acceptance is different. Architectural design has a relatively large tolerance for the result compared to general engineering products. Even if the performance of the building is unsatisfactory, the performance of the overall building can be improved by the resident adapting to or increasing the capacity of the plant system. In the case of a kinetic building envelope, however, unlike a typical building design, the tolerance to the design is low. The kinetic building envelope determines the sophisticated interactions between the components to determine the success of the system. This interaction is based on the integration of the functions and roles of the various components.

The prototyping process involves hypothesis setting—experiment planning—evaluation and verification to identify and address the failure factors accurately. In this respect, the prototyping process is similar to the research process. Eliminating or minimizing these failure factors plays an important role in understanding the validity of the technology to be introduced in advance and in improving accuracy. The prototyping process is an integrated design process. In this paper, a prototyping process is proposed based on case studies. These cases are the result of a two-year (the second semester in 2014, the second semester in 2015) digital design studio. These results were analyzed by focusing on the complement of the digital and physical models and the role of the traditional design medium.

Various factors of the design environment should be considered so that the prototyping process can be applied to the practical process. For example, the resources in the work environment, the nature of the designer, the nature of the design tool, and the rules or customary behavior within the organization should be considered. The prototyping process of this study considers the influence of the resources of the educational environment and the characteristics of the designer, but there are few cases for comparing and applying the environment and characteristics of the practitioner to apply these to the work environment. The future prototyping process should be conducted not only for the students, but also for the working designers. In addition, it should be improved and made into a process that closely examines the conditions of the work environment by conducting a detailed analysis of the process of the design company or conducting surveys among the practitioners.

**Funding:** This research was supported by a grant (19AUDP-B127891-03) from the Architecture & Urban Development Research Program funded by the Ministry of Land, Infrastructure, and Transport of the South Korean government.

**Acknowledgments:** This research was supported by a grant (19AUDP-B127891-03) from the Architecture & Urban Development Research Program funded by the Ministry of Land, Infrastructure, and Transport of the South Korean government.

**Conflicts of Interest:** The author declares no conflict of interest. The funders had no role in the design of the study; in the collection, analyses, or interpretation of data; in the writing of the manuscript, or in the decision to publish the results.

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
