# Peer review of "A Design Methodology Using Prototyping Based on the Digital-Physical Models in the Architectural Design Process"

_sustainability, doi:10.3390/su11164416_

Round 1

Reviewer 1 Report

An interesting paper addressing an important topic. The paper makes a good argument for the benefits of prototyping using a mix of physical and digital models as an alternative design method for facades. The paper offers a good review of the topic and presents some interesting case studies. But the main focus of the paper is work that seems to have taken place in one design studio that utilized this approach. The studio is the evaluated mostly through the instructor’s personal assessment as well as the use of pre and post surveys of student impressions of the topic.

My main concern is that I am not sure if this limited amount of original work (one design studio) and assessment can provide a sufficient bases for the generalized claims that the paper is making. I suggest reframing the paper in the context of the original work being presented as one case study testing this proposed model, and refraining from making any generalized claims about the effectiveness of the proposed model as the current work does not support such generalizations. Also, the paper needs to clearly distinguish between the description of the original work performed and the general descriptions of the potentials that the proposed approach can have (for example, the paper at several points references environmental performance, yet the work in the studio did not seem to address this issue, same with regard to the focus on façade design vs. other aspects of building design). The original work must be clearly identified and distinguished form literature review and/or theoretical claims. At several points within the paper, it was not easy to distinguish the two.

Additionally, I think the paper needs to be much clearer about the limited scope. The abstract did not really make that clear, and it should. Similarly, the research scope and methodologies section should clearly state that this work is based only on one educational design studio.

Other suggestions include the following:

-       Reduce the length of the abstract, make it a lot more concise, and make sure it clearly describes the scope and methods used. The current abstract reads more like an introduction.

-       Both the abstract and the research background sections make repeated claims about the complementarity of digital and physical models. While I do not necessarily disagree with that, the author needs to make a stronger case for that within the background section since it is a main premise for the work.

-       The research scope and methodologies section needs to be reworked int h context of the reduced scope and needed clarity of methods discussed above. At a minimum, much more detail needs to be provided about the design studio in question (educational level, number of students, scope, etc.), and the assessment methods used. 

-       The term “performance” needs to be clearly defined, both in general and in terms of what performances did the actual studio work address.

-       The issue of the low generalizability of findings should be clearly discussed. While I think the proposed work is interesting, it still doesn’t allow for considerable generalizability.

-       A plan for future work could be discussed to discuss the methodological limitations of the current work.

Author Response

Despite your busy schedule, thanks for your quick review of my paper. Sorry for my late feedback. The paper has been revised as follows. Please review it again.

1. Reduce the length of the abstract - pp.1: abstract

2. Make a stronger case for that within the background section since it is a main premise for the work.

 - pp. 4:140~165

3. The research scope and methodologies section needs to be reworked int h context of the reduced scope and needed clarity of methods discussed above. At a minimum, much more detail needs to be provided about the design studio in question (educational level, number of students, scope, etc.), and the assessment methods used.

 - line 127 ~ 165, 187 ~217

4. The term “performance” needs to be clearly defined, both in general and in terms of what performances did the actual studio work address.

- pp.2: footnote "performance"

5.  The issue of the low generalizability of findings should be clearly discussed. While I think the proposed work is interesting, it still doesn’t allow for considerable generalizability. 

- line 271 ~286

6. A plan for future work could be discussed to discuss the methodological limitations of the current work.

- line 748~758

Reviewer 2 Report

The author dealt with a very interesting topic - methodology to designing smart building envelopes. The article is very scientific, brings a lot of new knowledge, but very hard, complicated to reader.The article has many inaccuracies. I only focused on some:

The abstract is very long (426 words). I would recommend shortening.

Numbering of used literature - literary sources could be numbered from 1,2,3, ... to last (sometimes an exception may occur ...).

Explain the meaning of sentences on lines 116-118.

The development of the research is nicely illustrated in Figure 1.

Draw tables 1,2  according to regulations. Draw all tables according to template.

On line 467 is Figure 1, not accidentally Figure 5? Similarly, on line 501, is table 1 again? In my opinion, between lines 501 and 502 are pictures (or one picture with parts), not a table! Figures 2,3,4, describe what is on the X axis and the Y axis! ....to line 546...

Figure 1 is located on line 182, Figure 1 is also located on line 467?

Figure 2 is located on line 252, Figure 2 is also located on line 539?

Figure 3 is located on line 322, Figure 3 is also located on line 534?

Figure 4 is located on line 437, Figure 4 is also located on line 546?

Enter values in Figure on line 539 - Important factors in designing SBEs!

Figure 5 on line 596 is completely opaque, incomprehensible. What did the author say? What do those circles mean?!

The texts focus on hypotheses, facts, failures. They are hard to understand. They are hard to imagine the facts. These are futuristic scenes, examples, situations.

I leave it to the editor to edit the article to make it more readable to the ordinary reader.

Author Response

Despite your busy schedule, thanks for your quick review of my paper. Sorry for my late feedback. The paper has been revised as follows. Please review it again.

1. Reduce the length of the abstract - pp.1: abstract

2. Make a stronger case for that within the background section since it is a main premise for the work.

 - pp. 4:140~165

3. The research scope and methodologies section needs to be reworked int h context of the reduced scope and needed clarity of methods discussed above. At a minimum, much more detail needs to be provided about the design studio in question (educational level, number of students, scope, etc.), and the assessment methods used.

 - line 127 ~ 165, 187 ~217

4. The term “performance” needs to be clearly defined, both in general and in terms of what performances did the actual studio work address.

- pp.2: footnote "performance"

5.  The issue of the low generalizability of findings should be clearly discussed. While I think the proposed work is interesting, it still doesn’t allow for considerable generalizability. 

- line 271 ~286

6. A plan for future work could be discussed to discuss the methodological limitations of the current work.

- line 748~758

7. Check figures and tables

- Modified figures and tables

Reviewer 3 Report

This paper focused on a synergistic effect of both digital and physical model based prototyping in the case of architectural design. Questions and comments for the author would go like this:

Abstract part of the paper needs to be recomposed to be more brief and succinct for representing essential findings or differentiation points of the study carried out in the field of architectural design methodology. Current version of abstract looks a simple summary of the entire draft.

    2. All Figures and Tables are to be briefly explained what they are in the main draft.

    3. The original sources for case studies, referenced projects at different architectural design firms, school studios and other test cases need to be clarified along with the way the author collected and analysed such 'real-world' data sets. For instance, data sources and validity of the cases included in Table 1, 2, and 3(mistyped as Table 1) should be described.

     4. The most serious conjecture the author made through a series of arguments in the draft would be the effectiveness of 'Digital-Physical Model' based prototyping process in architectural design. And yet, both physical and virtual/conceptual models have long been used  in architectural design; only recently we witnessed the emergence of digital tools and models with the advance of computation. Hand drawn sketches and diagrams have been more powerful for concept generation than digital models. It seems that the author need to suggest more concrete and systematic framework how 'digital model' and 'physical model' can be integrated to provide a more powerful methodology in architectural design process instead of using them separately for a set of specific purposes. 

   5. Unlike 'physical model', 'digital model' could be in a variety of forms. Sophisticated 3 D modelers, VR, AR, animations and multi-purpose 3-D based engineering simulators such as SolidWorks enlightens a new horizon of representing Form, Material, Operation and Fabrication aspect of an architectural design. Author should include those newly emerging 'digital model' tools as potential replacement or partners of 'physical models' in the design process. 

Author Response

Despite your busy schedule, thanks for your quick review of my paper. Sorry for my late feedback. The paper has been revised as follows. Please review it again.

1. Reduce the length of the abstract - pp.1: abstract

2. Make a stronger case for that within the background section since it is a main premise for the work.

 - pp. 4:140~165

3. The research scope and methodologies

 - line 127 ~ 165, 187 ~217

4.  The issue of the low generalizability of findings

- line 271 ~286

5. A plan for future work

- line 748~758

6. Uses of sketches

In this paper, sketch is considered as important raw data for observing the result of design. It is one of the clues collected during the experiment. However, because the levels vary widely according to students' hand sketching abilities, it is true that they remain mysterious about how to analyze sketches. This will be analyzed later through other research themes.

Round 2

Reviewer 1 Report

The paper shows improvement from the previous submission especially with regard to clarity of methods. Issues with low generalizability of findings still persist. However, an argument is presented that the paper follows a qualitative methods approach, which is fine but needs more clarification.  

The second draft does discuss two methods of analyzing the work of the students, surveys and video recordings. The mention of video recordings, I believe, is a new addition that was not included in the previous version. However, the results and conclusions section do not address the video recordings? nor do they discuss how those recordings informed the outcomes? Addressing that as well as any consistencies or differences between the he two methods will add to the quality of the paper. 

The narrowing down of the focus form the more general performance to a more focused discussion of productivity is also good, and much clearer than the previous version. A focus of production also shows in the case studies section. However, the term production never shows up in the discussion of the actual case study the paper is based on, the design studio in question, nor in the findings of the paper.  I suggest carrying that focus on production through all parts of the paper especially the findings

The paper includes some types and is in need of careful editing. For example, line 127 states:" The purpose of this methodology is to increase the productivity of architects who are fundamentally faced by architects.", which is somewhat meaningless. 

Author Response

Thank you for your valuable review though you are busy.
The paper has been supplemented by adding the following:
1. Relationship between performance and productivity (pp.3:136-4:143)
2. Description of the purpose and results of video recording(pp.19:676 - 20:688)

Reviewer 2 Report

In this study, a prototype-based design methodology is proposed. The aim is to improve the functionality of the test and distinguish it from the prototypes that are implemented in conventional architectural design projects. The aim is also to explore reference cases that allow designers to maximize the use of digital models and physical models currently used in architectural designs. It is also about exploring the complementary roles and effects of digital and physical models. It is one of the most challenging topics in architectural design and requires innovative design, including testing and risk management. A conceptual prototype model that reflects the subject is applied to the design studio. Implementing this method requires strengthening validations and improving the accountability of ideas in architectural design phases. The design methodology allows designers to apply materials and manufacturing methods using digital and physical models. The topic is appropriately processed, all comments from the reviewer are incorporated. I recommend publishing as it is.

Author Response

(The authors gave the same response as above.)

Reviewer 3 Report

Author's revision of previous draft seems to be  quite substantial across all sections in the paper.

Even though the issues of digital vs. physical modeling in architectural design are controversial and difficult for systematic analysis, this study at least shows a design methodological investigation into two major design representation techniques. Further interesting researches on this specific issue is expected.  

Author Response

(The authors gave the same response as above.)
